# Epidemiology of the SARS-CoV-2 Omicron Variant Emergence in the Southeast Brazilian Population

**DOI:** 10.3390/microorganisms12030449

**Published:** 2024-02-23

**Authors:** Evandra Strazza Rodrigues, Svetoslav Nanev Slavov, Debora Glenda Lima de La Roque, Elaine Vieira Santos, Josiane Serrano Borges, Mariane Evaristo, Péricles Natan Mendes da Costa, Juliana de Matos Maçonetto, Adriana Aparecida Marques, Anemarie Dinarte Baccarin, Renata Aparecida Machado Oliveira, Wilson Lau Junior, Bruno Iglesias Benincasa, Luana Martins de Andrade da Cruz, Alex Ranieri Jerônimo Lima, Gabriela Ribeiro, Vincent Louis Viala, Loyze Paola Oliveira de Lima, Antonio Jorge Martins, Claudia Renata dos Santos Barros, Elaine Cristina Marqueze, Jardelina de Souza Todao Bernardino, Rejane Maria Tommasini Grotto, Jayme A. Souza-Neto, Vagner Fonseca, Maurício Lacerda Nogueira, Heidge Fukumasu, Luiz Lehmann Coutinho, Rodrigo Tocantins Calado, Dimas Tadeu Covas, Marta Giovanetti, Luiz Carlos Junior Alcantara, Sandra Coccuzzo Sampaio, Maria Carolina Elias, Simone Kashima

**Affiliations:** 1Blood Center of Ribeirão Preto, Ribeirão Preto Medical School, University of São Paulo, Ribeirão Preto 14051-140, SP, Brazil; svetoslav.slavov@hemocentro.fmrp.usp.br (S.N.S.); debora.laroque@usp.br (D.G.L.d.L.R.); joseane.borges@hemocentro.fmrp.usp.br (J.S.B.); mariane.evaristo@hemocentro.fmrp.usp.br (M.E.); pnmc.biotec@gmail.com (P.N.M.d.C.); jumatos.m@gmail.com (J.d.M.M.); adriana@hemocentro.fmrp.usp.br (A.A.M.); anedinarte@hemocentro.fmrp.usp.br (A.D.B.); rtcalado@usp.br (R.T.C.); dimas@fmrp.usp.br (D.T.C.); skashima@hemocentro.fmrp.usp.br (S.K.); 2Butantan Institute, São Paulo 05585-000, SP, Brazil; elainevs@alumni.usp.br (E.V.S.); renata.machado@hemocentro.fmrp.usp.br (R.A.M.O.); wilson.junior@hemocentro.fmrp.usp.br (W.L.J.); bruno.benincasa@hemocentro.fmrp.usp.br (B.I.B.); luana.andrade@hemocentro.fmrp.usp.br (L.M.d.A.d.C.); alex.lima@butantan.gov.br (A.R.J.L.); gabriela.rribeiro@butantan.gov.br (G.R.); vincent.viala@fundacaobutantan.org.br (V.L.V.); loyze.lima@butantan.gov.br (L.P.O.d.L.); antonio.martins@butantan.gov.br (A.J.M.); claudia.barros@butantan.gov.br (C.R.d.S.B.); elaine.marqueze@butantan.gov.br (E.C.M.); jardelina.bernardino@butantan.gov.br (J.d.S.T.B.); sandra.coccuzzo@butantan.gov.br (S.C.S.); carolina.eliassabbaga@butantan.gov.br (M.C.E.); 3School of Agricultural Sciences, São Paulo State University (UNESP), Botucatu 18610-034, SP, Brazil; rejane.grotto@unesp.br; 4Department of Diagnostic Medicine/Pathobiology, Kansas State Veterinary Diagnostic Laboratory College of Veterinary Medicine, Kansas State University, Manhattan, KS 66506, USA; jayme.souza-neto@unesp.br; 5Organização Pan-Americana da Saúde/Organização Mundial da Saúde, Brasília 70075-900, DF, Brazil; vagner.fonseca@gmail.com; 6Medicine School of São José do Rio Preto (FAMERP), São José do Rio Preto 15090-000, SP, Brazil; mauricio.nogueira@edu.famerp.br; 7Department of Veterinary Medicine, School of Animal Science and Food Engineering, University of São Paulo, Pirassununga 05508-270, SP, Brazil; fukumasu@usp.br; 8Centro de Genômica Funcional da ESALQ, University of São Paulo, Piracicaba 13418-900, SP, Brazil; llcoutinho@usp.br; 9Laboratorio de Flavivirus, Fundação Oswaldo Cruz, Rio de Janeiro 21040-360, RJ, Brazil; marta.giovanetti@ioc.fiocruz.br (M.G.); alcantaraluiz42@gmail.com (L.C.J.A.); 10Department of Science and Technology for Humans and the Environment, University of Campus Bio-Medico di Roma, 00128 Rome, Italy

**Keywords:** SARS-CoV-2 genomic surveillance, Brazilian population, Omicron variant

## Abstract

The aim of this study was to describe epidemiological characteristics and perform SARS-CoV-2 genomic surveillance in the southeastern region of São Paulo State. During the first months of 2022, we compared weekly SARS-CoV-2 infection prevalence considering age, Ct value, and variants’ lineages. An increase in the number of SARS-CoV-2-positive cases until the fourth epidemiological week of 2022 was observed. From the fourth epidemiological week onwards, the number of tests for SARS-CoV-2 diagnosis began to decrease, but the number of positive samples for SARS-CoV-2 remained high, reaching its most expressive level with a rate of 60% of infected individual cases. In this period, we observed a progressive increase in SARS-CoV-2 infection within the 0–10 age group throughout the epidemiological weeks, from 2.8% in the first epidemiological week to 9.2% in the eighth epidemiological week of 2022. We further observed significantly higher Ct values within younger patient samples compared to other older age groups. According to lineage assignment, SARS-CoV-2 (BA.1) was the most prevalent (74.5%) in the younger group, followed by BA.1.1 (23%), BA.2 (1.7%), and Delta (1%). Phylogenetic analysis showed that BA.2 sequences clustered together, indicating sustained transmission of this Omicron VOC sub-lineage by that time. Our results suggest the initial dissemination steps of the Omicron’s sub-linage BA.2 into the younger group, due to specific genomic features of the detected sequences. These data provide interesting results related to the spread, emergence, and evolution of the Omicron variant in the southeast Brazilian population.

## 1. Introduction

Although years have passed since the description of the first cases of SARS-CoV-2 infection in Wuhan, China, COVID-19 remains active but without considerable morbidity. SARS-CoV-2-infected individuals in general develop a severe acute respiratory syndrome; however, neurological [1], cardiovascular, cutaneous, and renal manifestations [2,3], and multiple organ system disorders [4], are also observed. Most individuals with COVID-19 develop a mild respiratory illness, such as a dry cough, fever, dyspnea, and asymptomatic cases. However, the severity of COVID-19 clinical symptoms seems to be related to different factors, such as the period of virus exposure and consequent viral load, host immune capacity, population immunization rate, age, and characteristics of the circulant SARS-CoV-2 variant [5].

In Brazil, until January 2022, three epidemic waves have been observed, driven by different variants of SARS-CoV-2. The first wave occurred in early April 2020 and was triggered by the strains B.1.1.28 and B.1.1.33 [6]. The second and most devastating epidemic wave occurred between December 2020 and March 2021 and was marked by thousands of fatal cases due to the introduction of the P.1 (Gamma) strain [7,8,9], followed by Delta SARS-CoV-2 variants of concern (VOC). In December 2021, the number of positive cases for SARS-CoV-2 increased again, due to the rapid emergence of the Omicron variant [10].

The Omicron variant’s emergence brought a worldwide concern related to the number of positive cases of SARS-CoV-2 [11,12]. The first description of the Omicron lineage variant was reported in southern Africa by November 2021 [11] and, in the same month, was detected in Brazil, in the state of São Paulo. During January 2022, the Omicron lineage was the dominant strain in Brazil and its introduction caused a sharp increase in the number of COVID-19 cases [10]. Some studies have demonstrated the circulation of the Omicron variant also in specific regions, such as Mato Grosso do Sul [13], Pará [14], Sergipe [15], Maranhão [16], Amazonas [17], and Minas Gerais [18]. According to the literature, the rapid spread of Omicron is attributed to the presence of multiple mutations in the spike protein that may confer greater binding affinity to the host cell receptor and an increased ability to evade immunity induced by vaccination and previous infection. 

Although Omicron causes milder symptoms and less clinical severity in vaccinated individuals compared to previous SARS-CoV-2 variants, there are public health concerns due to the large number of cases, uncertainty regarding the effectiveness of the vaccines, and especially because in some countries, children were not eligible for vaccination or boosters during the Omicron wave period [19]. Even a milder infection can lead to an uncontrolled number of infections, which, in turn, will cause an overload of healthcare systems. Furthermore, observations regarding the reduced severity of clinical disease induced by Omicron may not accurately reflect the virulence of this variant, as significant mortality due to Omicron has been observed in elderly and under-vaccinated populations [20]. Omicron was also associated with an increased risk of maternal morbidity and severe complications, mostly among symptomatic and unvaccinated women [21]. 

In this sense, to understand the epidemiological characteristics during the Omicron variant wave, we investigated, through genomic surveillance, the frequency, introduction, and viral load in children, young people, and elderly people infected by SARS-CoV-2 in the state of São Paulo. 

## 2. Materials and Methods

### 2.1. Sample Collection

Nasopharyngeal swabs were collected from individuals presenting with acute respiratory symptoms (fever, cough, sore throat, rhinorrhea, myalgia, headaches, anosmia, ageusia, and fatigue) in the Hospital das Clínicas of the Medical School of Ribeirão Preto/USP and State Health Departments. After that, the samples were received and processed by the COVID-RP Laboratory of the Blood Center of Ribeirão Preto to perform the molecular SARS-CoV-2 diagnostic.

The samples were collected during the first 10 epidemiological weeks of 2022 (2 January 2022 to 12 March 2022). A total of 43,206 individuals from the region of southeast Brazil were submitted to molecular diagnostics of SARS-CoV-2, and incidence of SARS-CoV-2-positive samples was monitored month by month, including the Ct value and demographic profile of the patients. The resampling consisted of 87 (0.5%) of the patients. In addition, we performed genome analysis in 3099 samples. 

### 2.2. RNA Extraction and SARS-CoV-2 Infection Diagnosis

For RNA extraction, we used the Extracta kit fast–DNA and RNA (Loccus do Brasil Ltd., Cotia, SP, Brazil) in the automatic extractor Extracta 32 (Loccus do Brasil Ltd., SP, Brazil) according to the manufacturer’s instructions. The SARS-CoV-2 infection diagnosis was performed using the COVID-19 Plus RealAmp Gene Finder^TM^ kit (OSANG Healthcare Co. Anyang, Gyeonggi, Republic of Korea). This assay allows for the molecular detection of the ORF1ab gene encoding RNA-dependent RNA polymerase (RdRp), the envelope (E) gene, and the nucleocapsid (N) gene of the SARS-CoV-2 virus, as well as an internal control by reverse transcription and real-time multiplex PCR. The mixture of 20 μL PCR final reaction volume contained 10 μL of COVID-19 Plus Reaction Mix, 5 μL of COVID-19 Plus Probe Mixture, and 5 μL of sample RNA. The amplification reaction was performed on an ABI Prism 7500 Sequence Detection System (Thermo Fisher Scientific, San Diego, CA, USA). The thermocycling conditions consisted of reverse transcription at 50 °C for 20 min, followed by denaturation at 95 °C for 5 min and 45 cycles of 95 °C for 15 s and 58 °C for 60 s. Fluorescence was read in the FAM, VIC, Texas Red, and Cy5 channels, and data analysis, including threshold values (cycle threshold (Ct), baseline start, final values, and interpretation of results, was defined according to the manufacturer’s instructions. According to the manufacturer, all samples with cycle threshold (Ct) amplification values ≤ 35 for the internal control were considered for the interpretation of the results. Samples with amplification for the RdRp, E, and N genes of SARS-CoV-2 exhibiting Ct ≤ 40 were considered positive, except for the E gene alone. All samples with Ct ≤ 35 for viral genes were selected for the SARS-CoV-2 genomic sequencing. 

### 2.3. SARS-CoV-2 Genomic Sequencing

The SARS-CoV-2 genomic libraries were generated using the COVIDSeq kit (Illumina, San Diego, CA, USA) [22] and Oxford Nanopore Sequencing Kit (SQL-LSK 109) [23,24] following the manufacturer’s specifications. All samples with Ct ≤ 35 for at least two viral genes were selected for the SARS-CoV-2 genomic sequencing.

Briefly, the RNA (500–4000 ng) was incubated with a specific primary buffer for 3 min, followed by cDNA synthesis. The generated cDNA was subjected to a multiplex PCR reaction using an ARCTIC v3 primer set, available at the GitHub repository (https://github.com/artic-network/artic-ncov2019/blob/master/primer_schemes/nCoV-2019/V3/nCoV-2019.tsv, accessed on 1 October 2021).

The PCR reaction products were combined for the tagmentation step of the genomic library. The tagged amplicon was purified and then amplified using indexes. After this point, the reactions were purified with the aid of magnetic beads, and the products from the libraries were grouped, forming a single genomic library. We proceeded with library purification using 1x AMPure XP Beads (Beckman Coulter, Brea, CA, USA) following the manufacturer’s protocol. At the end of the process, the library was eluted in a specific buffer and the genetic material was quantified using Qubit® dsDNA HS DNA Quantitation Kits (Invitrogen, San Diego, CA, USA), following the manufacturer’s instructions. Paired-end libraries were sequenced on Illumina’s MiSeq (V2 kit, 2150 cycles) or NextSeq 2000 (P2 kit, 2100 cycles) platforms.

For Nanopore technology, amplicons were purified using 1× AMPure XP beads (Beckman Coulter, Brea, CA, USA), and cleaned-up PCR product concentrations were measured using the Qubit dsDNA HS assay kit. DNA library preparation was carried out using the ligation sequencing kit and the native barcoding kit (NBD104 and NBD114, Oxford Nanopore Technologies, Oxford, UK) [24,25]. Sequencing libraries were loaded into a FLO-MIN106 flow cell (Oxford Nanopore Technologies). In each sequencing run, we used negative controls to prevent and check for possible contamination with less than 2% mean coverage. 

### 2.4. Generation of Consensus Sequences

Illumina raw sequences were submitted to quality control analysis using the FastaQC software version 0.11.8. Cleaning was performed using Trimmomatic version 0.3.9 to select the best quality sequence score (>30). We mapped the quality-filtered sequences against the SARS-CoV-2 reference (Genbank RefSeq NC_045512.2) using BWA and used SAMtools for indexing the mapping results.

The mapped files were submitted for improvement using Pilon to correct possible deletions and insertions caused by the mapping process. The quality-filtered sequences were subjected to a remapping against the genome improved by Pilon. Finally, we used bcftools for variant calling and seqtk for the assembly of the consensus SARS-CoV-2 genomes. Positions covered by fewer than 10 reads (DP < 10) and bases with a quality score lower than 30 were considered as an assembly gap and thus converted into Ns. Coverage values for each genome were calculated using SAMtools v1.12. We assessed the consensus genome sequence quality using Nextclade v0.8.1.11.

Nanopore raw files were base-called using Guppy, and barcode demultiplexing was performed using qcat. Consensus sequences were generated by de novo assembling using Genome Detective (https://www.genomedetective.com/, accessed on 2 January 2022).

### 2.5. Phylogenetic Analysis

For phylogenetic analysis, we analyzed 471 newly obtained sequences against a dataset containing 4660 SARS-CoV-2 genomes, which were obtained from GISAID (https://www.gisaid.org/, accessed on 2 January 2022). For virus taxonomy, we followed the steps of an already established canonical pipeline, as implied by Nextstrain v.3.0.3. [25]. In brief, viral genomes were aligned using MAFFT v.7 and IQtree v.3 was used for phylogenetic tree reconstruction. The maximum likelihood (ML) phylogenetic trees were visualized using FigTree v. 1.4.4 and additionally edited by the R ggtree package (R package for the visualization of trees and annotation).

### 2.6. Statistical Analysis

The statistical analysis was performed in GraphPad Prism version 5.0 (La Jolla). We applied Kruskal–Wallis and Dunn tests for multiple comparisons. We considered α = 0.05, and **p* < 0.05, ** *p* < 0.01, and *** *p* < 0.001 were considered statistically significant. For comparison between Ct values and age, we considered the SARS-CoV-2 N gene due to its major sensibility in the qPCR reaction (5 copies/μL for RdRp and E gene and 2 copies/μL for N gene). 

## 3. Results

Initially, we performed an analysis to investigate the incidence of SARS-CoV-2-positive cases in our region in the first 10 epidemiological weeks of 2022. During this period, the positivity rate for SARS-CoV-2 reached 41% (17,695/43,216), gradually increasing month by month. Of these, 779 (4.40%) were from children of up to 10 years of age, 1633 (9.23%) from the adolescent population (11–19 years of age), 13,244 (74.85%) from the adult population (20 to 59 years of age), and 2039 (11.52%) from individuals > 60 years old (Appendix A). According to Figure 1, there was an increase of 1.2-fold in the number of tested SARS-CoV-2 samples from the 1st (2–8 January 2022) to the 3rd epidemiological week (16–22 January 2022). The increasing number of molecular tests was accompanied by an increase in the SARS-CoV-2 positivity, rising from 24% to 47%, respectively. From the 4th epidemiological week (23–29 January 2022), the number of SARS-CoV-2 tests decreased; however, the number of positive cases remained stable and reached high levels of 60% of the total number of infected individuals. From the 5th epidemiological week (30 January to 5 February 2022), the number of SARS-CoV-2 molecular exams continued to fall and the number of SARS-CoV-2-positive cases began gradually decreasing over the epidemiological weeks, reaching a 13% positivity rate for SARS-CoV-2 in the 10th epidemiological week (Figure 1).

Due to the increase in SARS-CoV-2 cases observed in the first months of the year 2022, we examined the demographic profile of the SARS-CoV-2-positive cases. Our main question was to understand whether SARS-CoV-2 was more prevalent according to age, since we considered that at the time of this work, the majority of the adult population, that is, those over 20 years old, would be immunized with both vaccine doses as well as the booster dose. As shown in Figure 2, the majority of SARS-CoV-2-positive cases occurred between the epidemiological weeks 4 and 5, and we also observed an increase in children of up to 10 years of age (from 2.8% in the 1st epidemiological week to 9.2% in the 8th epidemiological week; Figure 2). The average age of this group of children was 6 ± 3.2 years, with 48% female and 52% male. When we evaluated the same period for the adolescent population (11–19 years of age), we also observed an increase in the number of SARS-CoV-2-positive cases, rising from 9.8% to 14.8% in the 8th epidemiological week (Figure 2). In the adolescent population, the mean age was 17 ± 2.6 years, with 54% female and 46% male. The frequency of SARS-CoV-2-positive cases for the adult population (20 to 59 years of age) remained unchanged. This was also the case regarding adults over 60 years old, whereby we noted no significant difference in the number of SARS-CoV-2 infections over the epidemiological weeks (Figure 2). The increase in the number of pediatric SARS-CoV-2-infected individuals may be related to the emergence of a new variant or also due to the availability of vaccines, considering the immunization scheme adopted in Brazil for different ages (Figure 3A). 

To understand the relationship between the burden of SARS-CoV-2 and age group, we compared the amplification Ct values of the SARS-CoV-2 N gene obtained from individuals within different age groups. In the majority of the epidemiological weeks, we observed a significant difference in Ct values of samples belonging to younger patients compared to other age groups. The Ct values of samples from children aged 0–10 years were higher than the Ct values observed in individuals > 60 years old (Appendix A). According to our results, the younger age group presented higher Ct values for the viral SARS-CoV-2 N gene, suggesting a lower load in this group of patients (Figure 3B). Interestingly, when the pediatric group was divided by age into four groups, it was possible to observe a gradual decrease in the Ct value (higher viral load). Children aged up to 2 years had an average Ct of 29, and the age group from 3 to 5 years had an average Ct of 28. Along with the increase in age, the Ct value decreased: children aged 6 to 9 years old had a mean Ct of 27, and from 10 years old onwards the mean Ct was 26 (Figure 3B). For adults, we observed a gradual decrease, less evident than what was observed in the pediatric group (Appendix A). 

Subsequently, we investigated the circulating SARS-CoV-2 variants from a region of southeast Brazil, composed of 90 counties, including Araraquara, Barretos, Franca, and Ribeirão Preto, and together they reached a population coverage of 3,307,320 inhabitants. A total of 3099 whole genomes were generated, which were collected between 2 January and 12 March 2022. The mean age of sequenced SARS-CoV-2-positive individuals was 37 ± 18 years, with 1804 (58%) females and 1295 (42%) males. The data analysis revealed that SARS-CoV-2 sequencing achieved an average coverage of 98% of the gene and an average depth of 3414 (±1941) times per sample. In accordance with the epidemiological situation, the SARS-CoV-2 variants BA.1 (76.5%), BA.1.1 (21.5%), BA.2 (1.3%), and Delta (0.75%) were identified (Figure 4). Our analysis revealed the predominance of the Omicron lineage during the first 10 epidemiological weeks of the year 2022. The complete lineage substitution of Delta to Omicron lineage was observed after epidemiological week 3 (Figure 4). The first case of BA.2 was also identified in epidemiological week 4, and this variant was frequently detected until the end of the evaluated period (Figure 4). 

Considering the increased number of SARS-CoV-2 infections in younger patients, and that during the period of this study, they were only partially immunized, becoming an important source of viral dissemination, we investigated the distribution of the SARS-CoV-2 variants in this group. Demographic characteristics showed that the mean age of younger patients (0 to 19 years) was 14 ± 5.6 years, with 236 (50.1%) females and 235 (49.9%) males. Similar to adult (20 to 59 years of age) and older (>60 years old) groups, in younger patients, the Omicron lineage (BA.1) was the most prevalent (74.5%), followed by BA.1.1 (23%), BA.2 (1.7%), and Delta (1%) (Figure 5A). Then, we investigated the distribution of SARS-CoV-2 variants in these younger patients (0 to 19 years of age) considering four different Departments of Public Health (DPH) from São Paulo State (Araraquara, Barretos, Franca, and Ribeirão Preto). Our analysis revealed that in this group, 100% of the SARS-CoV-2 genomes from Araraquara (III) and Franca (VIII) DPH were classified as Omicron. A predominance of Omicron BA.1 was also observed in Barretos (V) (71%) and Ribeirão Preto (XIII) (75%) DPH (Figure 5B). The variant BA.1.1 appeared as the second variant well represented in our dataset, reaching 23% of all genomes. However, in Barretos, 1% of the sequence was classified as BA.2, and Ribeirão Preto showed the presence of BA.2 (2.3%) and Delta (1.7%) (Figure 5B). 

The phylogenetic analysis (Figure 5B) showed that almost all analyzed sequences (n = 458/471) belonged to the Omicron VOC, and a small number (n = 5/471) to Delta VOC and the BA.2 sub-lineage (n = 8/471). The small number of Delta VOC sequences corresponded to the gradual replacement of the Delta VOC in Brazil by the Omicron VOC. The Omicron VOC sequences BA.1. and BA.1.1 that were obtained from younger patients formed several small monophyletic clusters, which were randomly interspersed across all the Omicron VOCs circulating worldwide. This clustering might be related to several independent BA.1 and BA.1.1 introductions in this part of the São Paulo State, which corresponded to the epidemiological situation and the rapid dissemination of the Omicron VOC in the state of São Paulo by that time. We were also able to identify a small number of BA.2 sequences that were clustered together, probably indicating sustained transmission of this Omicron VOC sub-lineage by that time. However, due to the very initial steps of BA.2 introduction, we suggest that this clustering reflects the initial dissemination steps of this VOC in this part of the State, rather than specific genomic features of the detected BA.2 sequences. 

## 4. Discussion

Genomic surveillance of SARS-CoV-2 has allowed the monitoring of the infection, as well as the knowledge to follow the evolution of the SARS-CoV-2 variants worldwide. In this work, we monitored the SARS-CoV-2 infection in people living in southeastern Brazil during the first 10 weeks of the year of 2022. We observed a progressive increase in the number of SARS-CoV-2-positive cases until the 3rd epidemiological week. According to our results, this increase in infected individuals was directly related to the introduction of the Omicron variant in this region, since the phylogenetic analysis revealed that 76% and 22% of the SARS-CoV-2 genomes evaluated in this period were classified as BA.1 and BA.1.1, respectively. Other studies corroborate our finding, showing that the arrival of the Omicron VOC caused an important change in the epidemiological scenario, culminating in the replacement of the Delta variant by Omicron in approximately 3.5 weeks, and reached a dominant plateau in different Brazilian capitals [26]. Furthermore, one of the relevant features of the Omicron variant is its ability to cause reinfection. This aspect has been reinforced by research that demonstrated low cross-neutralization against the Omicron variant from previous non-Omicron viral infection or two-dose mRNA vaccination [27].

When we considered the age of the SARS-CoV-2-positive sample, it was observed that the number of children and adolescents infected had increased over the epidemiological weeks. This observation was also reported by studies that evaluated the pandemic caused by SARS-CoV-2 in other countries and raised the hypothesis that this increase was due to the greater potential for transmission of the Omicron variant, less frequent use of face masks among children than adults, and low vaccination rate in the pediatric population, with only children aged 12 years and older being eligible for vaccination at the time of the study [28]. Another issue that we investigated was the relationship between age and SARS-CoV-2 viral load by analyzing the Ct (threshold cycle) of the nucleocapsid viral region (N gene). Significant differences were found related to the Ct value and age, suggesting that younger age presents a higher Ct value for the viral SARS-CoV-2 N gene, and probably that a lower pro-viral load should be observed in this group of patients. Three previous studies have described that those viral loads in children with asymptomatic SARS-CoV-2 infection or mild illness are comparable [29] or slightly lower than viral loads in adults with SARS-CoV-2 infection [30,31]. These findings are in line with Jia’s study in 2021, which showed that the cytokine profiles of children with mild symptoms of COVID-19 resemble those of healthy children, reflecting a low level of inflammation [32].

It is important to note that according to our data, the number of children and adolescents with COVID-19 is increasing, and this group may be responsible for the spread of new variants of SARS-CoV-2. During the period of this study, this particular group of pediatric individuals was not eligible or had not completed the anti-SARS-CoV-2 immunization (Figure 3A) [28]. In addition, schools returned to face-to-face classes and many Brazilian states eased security measures by allowing large events, and the mandatory use of masks was no longer in force. Our analysis found that in the pediatric group, 1.7% of the samples were classified as BA.2 sub-lineage. The emergence of the Omicron lineage and subvariants BA.1, BA.1.1, BA.2, BA.3, BA.4, BA.5, and BA.2.12.2 was reported in different continents [11,33]. The Omicron variant presents milder disease, but it is significantly more contagious and has caused more hospitalizations, especially in unvaccinated children younger than 5 years old and unvaccinated or incompletely vaccinated adults [34,35]. 

One of the limitations of our study is related to the lack of vaccination data or information related to previous infection status. Considering this limitation, the vaccination status was inferred based on data from the Brazilian Ministry of Health, which determined that during the study period, the COVID-19 vaccine was available for those over 12 years of age, and children under 11 years of age were unvaccinated or only partially vaccinated. However, we cannot guarantee that the increased occurrence of SARS-CoV-2 in the pediatric group is related to the lack of vaccination or the insufficient time to generate a potent immune response. Another limitation was the inability to investigate antibody levels (IgG/IgM). The study was focused on molecular diagnosis using RNA extracted from nasopharyngeal specimens and, unfortunately, no additional blood samples were collected for further investigations during the study period. 

Even though we did not obtain data related to the symptoms of the evaluated individuals, we considered that the high number of SARS-CoV-2 infections did not increase the number of deaths, suggesting that the vaccines offered low effectiveness against Omicron infection, but significant protection against disease severity, especially after the booster dose. According to Araujo da Silva, the distribution age of patients admitted to a hospital in Rio de Janeiro during the Omicron season showed more children between 0 and 2 years and fewer children older than 12 years compared with the previous 2 years [36]. Furthermore, it was observed that only two (3.3%) hospitalized patients had received the complete COVID-19 vaccine, and the majority of admitted cases of the disease occurred in children where the vaccine was not available for their respective ages or who did not receive two doses [36]. 

In conclusion, we should be aware of the variation in the number of SARS-CoV-2-infected individuals and mainly the affected individuals’ ages. This may represent an evolutionary pressure for immune escape and encourage the emergence of new variants. Even with the progression of the COVID-19 vaccination in the Brazilian population, constant genomic surveillance for SARS-CoV-2 strains remains essential to determine the role of viral evolution and monitor the spread of variants of concern.

## Figures and Tables

**Figure 1 microorganisms-12-00449-f001:**
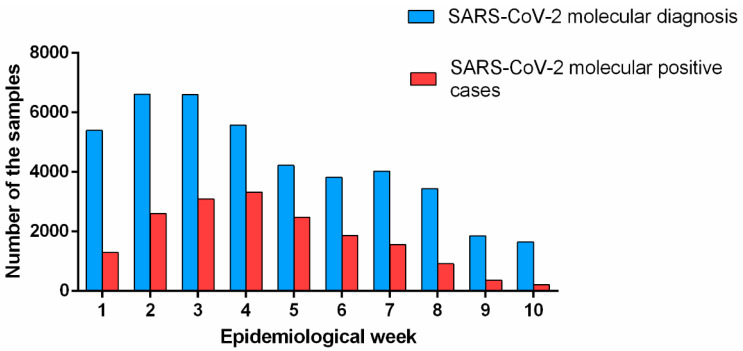
SARS-CoV-2 epidemiological dynamics in southeast Brazil. The blue bars show the number of samples tested with SARS-CoV-2 molecular diagnosis and the red bars show the number of samples positive for SARS-CoV-2 from 2 January to 12 March 2022 in southeast Brazil.

**Figure 2 microorganisms-12-00449-f002:**
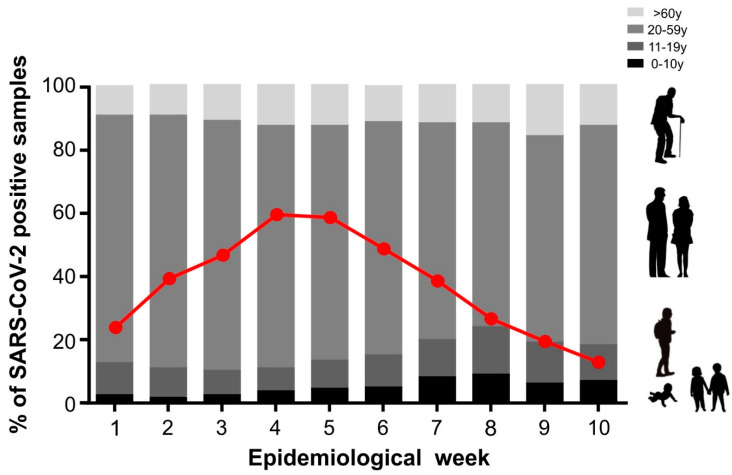
SARS-CoV-2-positive cases according to the age range throughout the epidemiological weeks. The different shades of gray represent the different ages. The red line represents the percentage of positive cases of SARS-CoV-2 by RT-PCR during the epidemiological weeks (1st to 10th).

**Figure 3 microorganisms-12-00449-f003:**
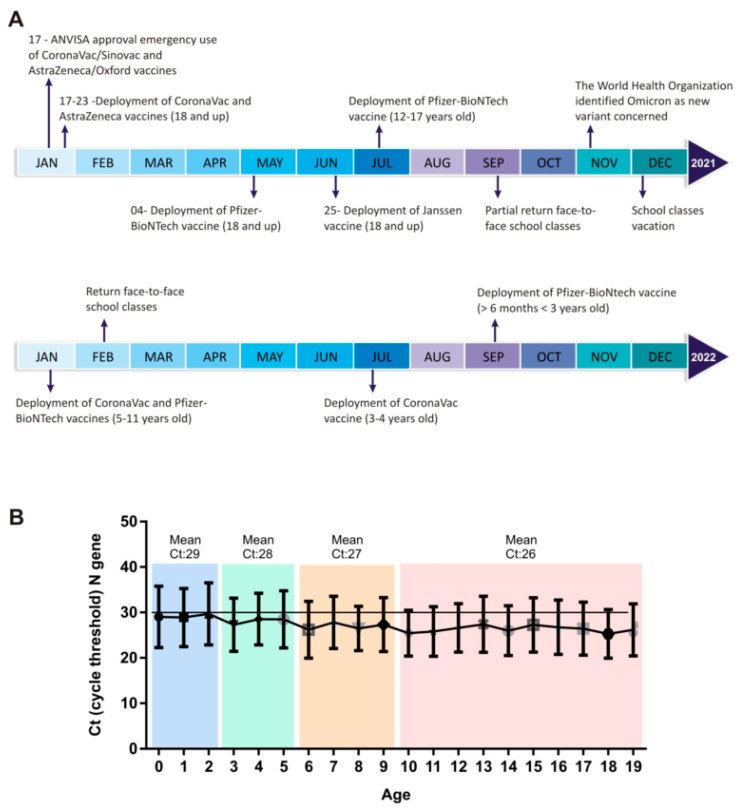
Timeline of key events related to the deployment of SARS-CoV-2 vaccination in Brazil (**A**). Real-time PCR analysis showing the Ct values for the N gene of SARS-CoV-2 (**B**). Each point means the mean (SD) of Ct values for the N gene of SARS-CoV-2 according to age. The line fixed at Ct = 30 facilitates comparative visualization. The different colors represent different clustered age groups.

**Figure 4 microorganisms-12-00449-f004:**
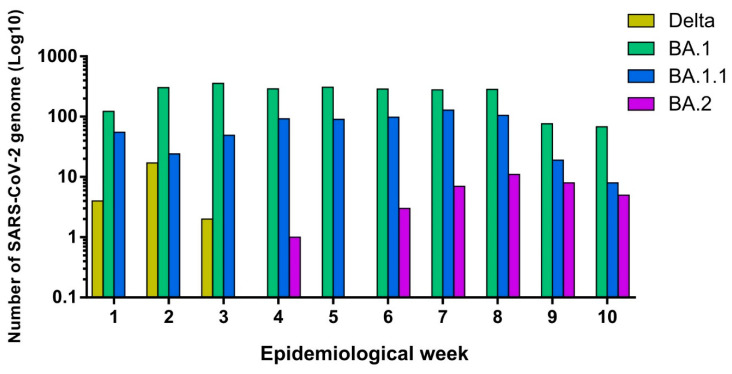
Graphic representation of the genetic and epidemiological profile of samples selected for SARS-CoV-2 genomic surveillance. Histograms showing the number of SARS-CoV-2 genomes that were detected. The color of the bars represents the different SARS-CoV-2 variants according to the epidemiological week.

**Figure 5 microorganisms-12-00449-f005:**
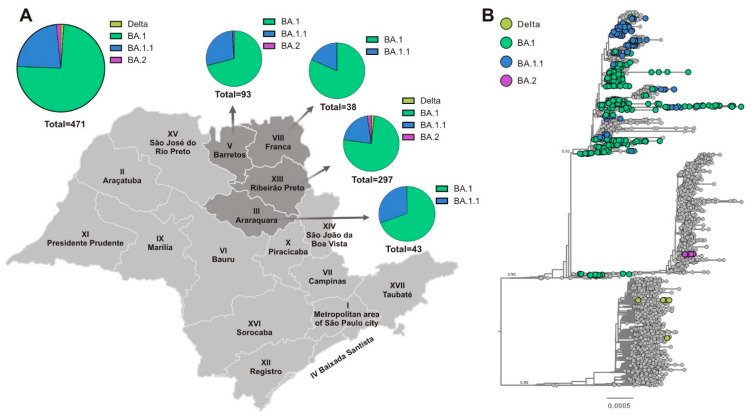
Evolution and spread of SARS-CoV-2 in people aged 0–19 years from southeast Brazil. (**A**) Distribution of SARS-CoV-2 genome variants in the total pediatric group and according to different regions from São Paulo State. (**B**) Phylogenetic analysis of the SARS-CoV-2 lineages detected in pediatric patients, including 471 new genomes generated in this study plus n = 4660 reference strains sampled worldwide, presented as a Maximum-Likelihood tree. Tips are colored according to the SARS-CoV-2 variant. Most of the samples were assigned as Omicron VOC (sub-lineages BA.1 and BA1.1), which coincided with the epidemiological situation within the State and Brazil. We also detected the initiation of the introduction of the BA.2 lineage in São Paulo State.

## Data Availability

The data are contained in the article or archived at Gisaid.org.

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
