# Peer review of "Epidemiology of the SARS-CoV-2 Omicron Variant Emergence in the Southeast Brazilian Population"

_microorganisms, 2024, doi:10.3390/microorganisms12030449_

Round 1

Reviewer 1 Report

Comments and Suggestions for Authors

Thank you for the opportunity of reviewing the manuscript. It is interesting and it reflects dedication and effort.

however, there are certain aspects to address in order to improve the manuscript.

In the abstract epidemiological weeks are mentioned, but not the year. In the text dates as expressed as dd-mm-yyyy please use months with wording or abreviared, as dates are differently expressed around the world. 

Introduction needs to be improved, as it seems generic and outdated.

One major area of improvement is that references are outdated, as Covid-19 has rapidly changed, almost all references are from 2022 or older, please update to include the most recent and reliable sources of information.

Comments on the Quality of English Language

Dates and places and also context description should to be written for a broader audience including English speaking audience 

Author Response

Reviewer #1:

“Thank you for the opportunity of reviewing the manuscript. It is interesting and it reflects dedication and effort. However, there are certain aspects to address in order to improve the manuscript”.

We thank Reviewer #1 for his/her valuable comments. We are grateful for the positive feedback of reviewer 1 on the presented manuscript. In order, to clarify the text we added improvements in the manuscript as described in the following.

1) “In the abstract epidemiological weeks are mentioned, but not the year. In the text dates as expressed as dd-mm-yyyy please use months with wording or abreviared, as dates are differently expressed around the world”.

We agree with Reviewer #1.  We revised all the text of the manuscript, included the year in the abstract section, and modified the form to express the data.  

Page 2, lines 90 and 91.  Now read as: “02 January 2022 to 12 March 2022”.

Page 4, lines 184 to 189.  Now read as:02-08 January 2022; 16-22 January 2022; 23-29 January 2022; 30 Jan. to 05 Feb. of 2022”

2) “Introduction needs to be improved, as it seems generic and outdated”.

In order to improve the introduction, we included the following modification on the text of the manuscript:

Page 2, lines 65 to 71. “The first description of Omicron lineage variant was reported in Southern Africa by November 2021 [11] and in the same month was detected in Brazil, in the state of São Paulo. During January 2022, Omicron lineage was the dominant strain in Brazil and its introduction caused a sharp increase in the number of COVID-19 cases [10]. Some studies have demonstrated the circulation of the Omicron also in specific regions such as Mato Grosso do Sul [13], Pará [14], Sergipe [15], Maranhão [16], Amazonas [17], and Minas Gerais [18]”.

3) “One major area of improvement is that references are outdated, as Covid-19 has rapidly changed, almost all references are from 2022 or older, please update to include the most recent and reliable sources of information”.

We comprehend that COVID-19 scenario changed rapidly became the reference outdated. We revised the manuscript, maintained the necessary references and update the bibliography according to the actual scientific reports.  

We included the following references:

Lamarca AP, Souza UJB, Moreira FRR, Almeida LGP, Menezes MT, Souza AB, et al. The Omicron Lineages BA.1 and BA.2 (Betacoronavirus SARS-CoV-2) Have Repeatedly Entered Brazil through a Single Dispersal Hub. Viruses. 2023 Mar 30;15(4):888. doi: 10.3390/v15040888. PMID: 37112869; PMCID: PMC10146814.

de Mello Almeida Maziero L, Giovanetti M, Fonseca V, Zardin MCSU, de Castro Lichs GG, de Rezende Romera GR, et al. Unveiling the Impact of the Omicron Variant: Insights from Genomic Surveillance in Mato Grosso do Sul, Midwest Brazil. Viruses. 2023 Jul 22;15(7):1604. doi: 10.3390/v15071604. PMID: 37515290; PMCID: PMC10386548.

Pinho CT, Vidal AF, Negri Rocha TC, Oliveira RRM, da Costa Barros MC, Closset L, et al. Transmission dynamics of SARS-CoV-2 variants in the Brazilian state of Pará. Front Public Health. 2023 Jul 5;11:1186463. doi: 10.3389/fpubh.2023.1186463. PMID: 37790714; PMCID: PMC10543262.

Freitas MTS, Sena LOC, Fukutani KF, Dos Santos CA, Neto FDCB, Ribeiro JS, et al. The increase in SARS-CoV-2 lineages during 2020-2022 in a state in the Brazilian Northeast is associated with a number of cases. Front Public Health. 2023 Dec 14;11:1222152. doi: 10.3389/fpubh.2023.1222152. PMID: 38186707; PMCID: PMC10771345.

de Sousa LAF, Ferreira LSS, Lobato LFL, Ferreira HLDS, Sousa LHDS, Santos VFD, et al. Molecular epidemiology of SARS-CoV-2 variants in circulation in the state of Maranhão, Brazil. J Med Virol. 2023 Sep;95(9):e29092. doi: 10.1002/jmv.29092. PMID: 37724346.

Arantes I, Bello G, Nascimento V, Souza V, da Silva A, Silva D, et al. Comparative epidemic expansion of SARS-CoV-2 variants Delta and Omicron in the Brazilian State of Amazonas. Nat Commun. 2023 Apr 11;14(1):2048. doi: 10.1038/s41467-023-37541-6. PMID: 37041143; PMCID: PMC10089528.

de Menezes MT, Moreira FRR, Whittaker C, Santos FM, Queiroz DC, Geddes V, et al. Dynamics of Early Establishment of SARS-CoV-2 VOC Omicron Lineages in Minas Gerais, Brazil. Viruses. 2023 Feb 20;15(2):585. doi: 10.3390/v15020585. PMID: 36851799; PMCID: PMC9962645.

Other modifications

We modified the conclusion including information: “Even with the progression of the COVID-19 vaccination in the Brazilian population, constant genomic surveillance for SARS-CoV-2 strains remains essential to determine the role of viral evolution and monitor the spread of variants of concern”.

Reviewer 2 Report

Comments and Suggestions for Authors

Dear Authors,

Congratulations on your work. I enjoyed reading your manuscript. However, before recommending publication it is important to make the following corrections/improvements for enhanced reading experience and flow -

a. Please use "," instead of "." when talking about numbers in thousands. For example, Line 91 - it should be 43,206 and not 43.206. Please make this correction across entire manuscript.

b. Please add the primer sequences, gene name, and amplicon length along with annealing temperatures, cycling conditions etc for the PCR part.

c. In section 2.2, the mentioned cycle cutoffs of 35 and 40 cycles, were they specified by the manufacturer or self considered by the author groups or derived from literature? More references and explanation of the cutoffs are needed for reader clarity.

d. Can the authors provide more information on the role of the 3 genes used in PCR - E, N, and RdRp and their importance in diagnostic testing of COVID? It seems that authors have presented results focused only on the N gene.

e. Can the author provide information that during the epidemiological weeks 1-10, which variant(s) of COVID-19 were generally more active/dominant in the country or the region - not from their analysis but from national epidemiological reports/data.

f. It would be worthwhile to present the data also on how many samples had N gene positive, E gene positive, RdRp gene positive and how many of those samples had 2 or more genes positive. I am aware authors mentioned that all 2+ gene positive samples were taken for genomic sequencing (3000+) but what I am looking for is in which combinations of those positive genes were present in those sequenced samples. Example, 200 were E+N positive, 500 were E+RdRp positive etc.

g. Do the authors have information about how many patients had previous infection with COVID-19? There is reported presumed information by authors based on vaccination age groups which also doesnt seems precise enough. The authors note that - "Our study compared the epidemiological data during the Omicron wave in the unvaccinated (under 5 years), partially vaccinated (5-11 years) and completely vaccinated (over 12 years)" - this is really not very indicative that all sampled participants were indeed un/partially/completely vaccinated just based on age groups. The authors show in Figure 3 that vaccination for 5-11 years old started in Jan 2022, and the study epidemiological period started in Feb 2022, so there is doubt that there was enough time for kids to generate a potent immune response from the vaccine. Hence, I am not very convinced on this point that authors try to make regarding vaccination and also on previous infection status (no information) since there can be differences. Antibody levels (IgG/IgM) will be needed to provide a better picture between vaccination/infection and viral loads. Authors should discuss this as a limitation that between vaccine deployment and their study sampling, there wasn't enough time and that its potential impact on results.

h. Were there any patients who were repeatedly sampled over the 10 weeks period of the study?

i. Figure 3B is quite interesting. Is it possible to extend it for all age groups or add more panels for adults?

j. It would be better to create a table with demographic characteristics like mean age and gender rather than split distribution/mention across the entire results section. It becomes difficult to follow and breaks the flow.

k. Authors should discuss the limitations of their study including logistical, sampling, and methodological.

Comments on the Quality of English Language

English language is fine.

Author Response

Reviewer #2:

Congratulations on your work. I enjoyed reading your manuscript. However, before recommending publication it is important to make the following corrections/improvements for enhanced reading experience and flow”.

We are very grateful for the positive feedback of reviewer #2. All the suggestions were considered and have permitted considerable improvement of the manuscript.  Thank you!

  1. “Please use "," instead of "." when talking about numbers in thousands. For example, Line 91 - it should be 43,206 and not 43.206. Please make this correction across entire manuscript”.

We thank the review #2 for this observation. All the numbers in thousands were modified as suggested.

  1. “Please add the primer sequences, gene name, and amplicon length along with annealing temperatures, cycling conditions etc for the PCR part”.

We agree with Reviewer #2 that the information about reagents and the conditions of PCR is important. In this study, we used the real-time PCR commercial kit COVID-19 Plus RealAmp Gene FinderTM kit (OSANG Healthcare Co. Ltd.). This molecular assay performs the one-step reverse transcription and real-time multiplex PCR for the qualitative detection of SARS-CoV-2 nucleic acids. The real-time PCR reaction amplified simultaneously the RdRp (RNA-dependent RNA polymerase) (FAM fluorescence), envelope protein (E) (Texas Red fluorescence) and nucleocapsid protein (N) (JOE fluorescence) genes of the SARS-CoV-2 virus and an internal control (Cy5 fluorescence). An internal control (IC), targeting the endogenous human RNase P gene is used to verify that nucleic acid is present in every sample and to ensure that samples resulting as negative for SARS-CoV-2 RNA contain nucleic acid for testing.

The following information was included in the manuscript:

Page 03, lines 116-118:

“This assay allows for the molecular detection of the ORF1ab gene encoding RNA-dependent RNA polymerase (RdRp), the envelope (E) gene, and the nucleocapsid (N) gene of the SARS-CoV-2 virus and an internal control by…”

  1. “In section 2.2, the mentioned cycle cutoffs of 35 and 40 cycles, were they specified by the manufacturer or self considered by the author groups or derived from literature? More references and explanation of the cutoffs are needed for reader clarity”.

The cycle threshold (Ct) cutoff was defined by the manufacturer´s fabricant. The reaction was considered valid for the interpretation of the results when the cycle threshold (Ct) amplification values ≤35 for the internal control, indicating that RNA extraction, reverse transcription, and real-time PCR work appropriately.  For the positive sample, we considered the amplification cycle threshold (Ct) cutoff Ct ≤40 for RdRp, E, and N genes of SARS-CoV-2. All the samples that present only amplification of the E gene were not considered positive for SARS-CoV-2, once the amplification region of the E gene is not specific for SARS-CoV-2 and can detect Sarbecovirus.

To clarify this point, we have added the following information to the manuscript:

Page 3, lines 128-129: “According to the manufacturer, all samples with cycle threshold (Ct) amplification values ≤35 …”

  1. “Can the authors provide more information on the role of the 3 genes used in PCR - E, N, and RdRp and their importance in diagnostic testing of COVID? It seems that authors have presented results focused only on the N gene”.

We agree with Reviewer #2 that is important to include more information about the reason for selecting the SARS-CoV-2 Nucleocapsid (N) Gene for our analysis. According to our results, the number of SARS-CoV-2 positive samples with the PCR reaction result unterminated or without an amplification curve was more frequent for RdRp and E genes of SARS-CoV-2 (Figure 01). These results corroborate the manufacturer's data, which describe the following analytical sensitivity for the Gene FinderTM kit: RdRp (5 copies/μL), N gene (2 copies/μL), and E gene (5 copies/μL). In this sense, the reason we selected the SARS-CoV-2 N gene for analysis is due to the manufacturer's recommendation and the results obtained in our data analysis.

Figure 1: Number of SARS-CoV-2 positive samples with the PCR reaction result unterminated or without an amplification curve in the qPCR analysis.

To clarify this point, we have added the following information to the manuscript:

Page 04, lines 190-193: For comparison between Ct values and age, we considered the SARS-CoV-2 N gene due to its major sensibility in the qPCR reaction (5 copies/μL for RdRp and E gene and 2 copies/μL for N gene).

  1. “Can the author provide information that during the epidemiological weeks 1-10, which variant(s) of COVID-19 were generally more active/dominant in the country or the region - not from their analysis but from national epidemiological reports/data”.

We thank Reviewer #2 for this observation. We investigated the introduction of Omicron lineage in different regions of Brazil, and to better discuss this point we included the following paragraph in the manuscript:

Page 02, lines 65 to 71

“The first description of Omicron lineage variant was reported in Southern Africa (Viana et al., 2022) and in the same month was detected in Brazil (30 November 2021), in the state of São Paulo, but other states quickly reported new cases. During January 2022, Omicron lineage was the dominant strain in Brazil and its introduction caused a sharp increase in the number of COVID-19 cases (Lamarca et al., 2023). Some studies have demonstrated the circulation of the Omicron also in specific regions such as Mato Grosso do Sul (de Mello et al., 2023), Pará (Pinho et al., 2023), Sergipe (Freitas et al., 2022), Maranhão (Sousa et al., 2023), Amazonas (Arantes et al., 2023) and Minas Gerais (de Menezes et al., 2023)”.

  1. “It would be worthwhile to present the data also on how many samples had N gene positive, E gene positive, RdRp gene positive and how many of those samples had 2 or more genes positive. I am aware authors mentioned that all 2+ gene positive samples were taken for genomic sequencing (3000+) but what I am looking for is in which combinations of those positive genes were present in those sequenced samples. Example, 200 were E+N positive, 500 were E+RdRp positive etc”.

According to the suggestion to review #2, we performed the analysis of the number of SARS-CoV-2 positive samples that presented the PCR reaction result unterminated or without an amplification curve for each evaluated gene. As mentioned in item “d” the SARS-CoV-2 Nucleocapsid (N) Gene has major sensibility in the qPCR reaction (2 copies/μL) when compared to RdRp and E gene (5 copies/μL), then the number of positive samples without amplification result of N gene is a rare event considering the large number of samples evaluated (Table 1). Further, we have to consider that the absence of detection of the SARS-CoV-2 Nucleocapsid (N) Gene in the positive sample can be attributed due the presence of N gene mutations, as reported before by Lesbon et al., 2021.

Table 1. The number of positive samples with the PCR reaction result unterminated considering the epidemiological week.

Regarding the genomic sequencing (3.099), we performed the analysis in samples with Ct ≤ 35 for viral genes, and considering this condition, all the samples presented amplification results for the three viral genes.

To clarify this point, we modified the following sentence on Page 03, line 132:

Before reading as “All samples with Ct ≤ 35 for at least two viral genes were selected for the SARS-CoV-2 genomic sequencing”

Now read as” All samples with Ct ≤ 35 for viral genes were selected for the SARS-CoV-2 genomic sequencing

  1. “Do the authors have information about how many patients had previous infection with COVID-19? There is reported presumed information by authors based on vaccination age groups which also doesnt seems precise enough. The authors note that - "Our study compared the epidemiological data during the Omicron wave in the unvaccinated (under 5 years), partially vaccinated (5-11 years) and completely vaccinated (over 12 years)" - this is really not very indicative that all sampled participants were indeed un/partially/completely vaccinated just based on age groups. The authors show in Figure 3 that vaccination for 5-11 years old started in Jan 2022, and the study epidemiological period started in Feb 2022, so there is doubt that there was enough time for kids to generate a potent immune response from the vaccine. Hence, I am not very convinced on this point that authors try to make regarding vaccination and also on previous infection status (no information) since there can be differences. Antibody levels (IgG/IgM) will be needed to provide a better picture between vaccination/infection and viral loads. Authors should discuss this as a limitation that between vaccine deployment and their study sampling, there wasn't enough time and that its potential impact on results”.

We agree with Reviewer #2 that data related to previous SARS-CoV-2 infections is important, but we don’t have access to this information in our cohort and have not performed this analysis. Another point that we consider relevant is the vaccination status. Although during the study period, the vaccine was available to adults, this does not guarantee that all adults were vaccinated. Furthermore, many individuals in the pediatric group may not have had the opportunity to be vaccinated or, for those vaccinated, they may not have the time to generate a potent immune response against SARS-CoV-2. To clarify this point, we modified the manuscript including this information as a limitation of our study.

Page 12, lines 365 to 373

“One of the limitations of our study is related to the lack of vaccination data or information related to previous infection status. Considering this limitation, the vaccination status was inferred based on data from the Brazilian Ministry of Health, which determined that during the study period, the COVID-19 vaccine was available for those over 12 years of age, and children under 11 years of age were unvaccinated or were partially vaccinated. However, we cannot guarantee that the increased occurrence of SARS-CoV-2 in the pediatric group is related to the lack of vaccination or the insufficient time to generate a potent immune response. Another limitation was the inability of antibody levels investigation (IgG/IgM). The study was focused on molecular diagnosis using RNA extracted from nasopharyngeal specimens, and unfortunately, no additional blood samples were collected for further investigations during the study period”.

  1. “Were there any patients who were repeatedly sampled over the 10 weeks period of the study”?

During the investigated period (January 1st to March 11th), we had 87 resampled patients. The average difference between collection days was 10 days (minimum 1 and maximum 46). In this circumstance, it is important to highlight the criteria usually considered to identify reinfections: at least 90 days between the first infection and second infection or different SARS-CoV-2 variants identified in 45-89 days after the first infection. Otherwise, it could be considered a persistent infection. These reinfection cases are under investigation in another study of our group.

The following information was included in the manuscript:

Page 03, line 105: “During this period, 17,695 (40.95%) samples were SARS-CoV-2 positive, including 87 (0,5%) resampled patients. Of these, 779 (4.40%) were from …”

  1. “Figure 3B is quite interesting. Is it possible to extend it for all age groups or add more panels for adults”?

We thank the review#2 for this observation. Interestingly, when the Ct analysis was extended to adults, we also observed a gradual decrease in the Ct value, although for adults the gradual decrease was less evident than what was observed in the pediatric group. However, we consider this information relevant and decided to include it in our manuscript.

 The following information was included in the manuscript:

Page 08, lines 254-255: For adults, we observed a gradual decrease less evident than what was observed in the pediatric group (Supplementary Figure 02).    

Supplementary Figure 02. Real-time PCR analysis showing the CT values for the N gene of SARS-CoV-2.

  1. “It would be better to create a table with demographic characteristics like mean age and gender rather than split distribution/mention across the entire results section. It becomes difficult to follow and breaks the flow”.

In order to improve the manuscript, we included the supplementary table with the demographic characteristics of our cohort. Once, our study was divided into different sections (epidemiological investigation, genomic surveillance, and phylogenetic analysis), we intended to include the epidemiological data in the text to indicate the exact characteristics of the group used in each analysis. 

The following supplementary table was included:

Supplementary material. Epidemiological characteristics of evaluated samples.

  1. “Authors should discuss the limitations of their study including logistical, sampling, and methodological”.

We agree with Review#2, that information about the limitations of our study is relevant to improve the manuscript.

The following information was included in the manuscript:

Page 12, line 365 to 376

“One of the limitations of our study is related to the lack of vaccination data or information related to previous infection status. Considering this limitation, the vaccination status was inferred based on data from the Brazilian Ministry of Health, which determined that during the study period, the COVID-19 vaccine was available for those over 12 years of age, and children under 11 years of age were unvaccinated or were partially vaccinated. However, we cannot guarantee that the increased occurrence of SARS-CoV-2 in the pediatric group is related to the lack of vaccination or the insufficient time to generate a potent immune response. Another limitation was the inability of antibody levels investigation (IgG/IgM). The study was focused on molecular diagnosis using RNA extracted from nasopharyngeal specimens, and unfortunately, no additional blood samples were collected for further investigations during the study period”.

The following information was removed to the manuscript:

“Our study compared the epidemiological data during the Omicron wave in the unvac-cinated (under 5 years), partially vaccinated (5-11 years) and completely vaccinated (over 12 years)”.

References

Centers for Disease Control and Prevention. Investigative criteria for suspected cases of SARS-CoV-2 reinfection (ICR) Available at: https://stacks.cdc.gov/view/cdc/96072 Cited in 2024/01/18.

Lamarca AP, Souza UJB, Moreira FRR, Almeida LGP, Menezes MT, Souza AB, et al. The Omicron Lineages BA.1 and BA.2 (Betacoronavirus SARS-CoV-2) Have Repeatedly Entered Brazil through a Single Dispersal Hub. Viruses. 2023 Mar 30;15(4):888. doi: 10.3390/v15040888. PMID: 37112869; PMCID: PMC10146814.

de Mello Almeida Maziero L, Giovanetti M, Fonseca V, Zardin MCSU, de Castro Lichs GG, de Rezende Romera GR, et al. Unveiling the Impact of the Omicron Variant: Insights from Genomic Surveillance in Mato Grosso do Sul, Midwest Brazil. Viruses. 2023 Jul 22;15(7):1604. doi: 10.3390/v15071604. PMID: 37515290; PMCID: PMC10386548.

Pinho CT, Vidal AF, Negri Rocha TC, Oliveira RRM, da Costa Barros MC, Closset L, et al. Transmission dynamics of SARS-CoV-2 variants in the Brazilian state of Pará. Front Public Health. 2023 Jul 5;11:1186463. doi: 10.3389/fpubh.2023.1186463. PMID: 37790714; PMCID: PMC10543262.

Freitas MTS, Sena LOC, Fukutani KF, Dos Santos CA, Neto FDCB, Ribeiro JS, et al. The increase in SARS-CoV-2 lineages during 2020-2022 in a state in the Brazilian Northeast is associated with a number of cases. Front Public Health. 2023 Dec 14;11:1222152. doi: 10.3389/fpubh.2023.1222152. PMID: 38186707; PMCID: PMC10771345.

de Sousa LAF, Ferreira LSS, Lobato LFL, Ferreira HLDS, Sousa LHDS, Santos VFD, et al. Molecular epidemiology of SARS-CoV-2 variants in circulation in the state of Maranhão, Brazil. J Med Virol. 2023 Sep;95(9):e29092. doi: 10.1002/jmv.29092. PMID: 37724346.

Arantes I, Bello G, Nascimento V, Souza V, da Silva A, Silva D, et al. Comparative epidemic expansion of SARS-CoV-2 variants Delta and Omicron in the Brazilian State of Amazonas. Nat Commun. 2023 Apr 11;14(1):2048. doi: 10.1038/s41467-023-37541-6. PMID: 37041143; PMCID: PMC10089528.

de Menezes MT, Moreira FRR, Whittaker C, Santos FM, Queiroz DC, Geddes V, et al. Dynamics of Early Establishment of SARS-CoV-2 VOC Omicron Lineages in Minas Gerais, Brazil. Viruses. 2023 Feb 20;15(2):585. doi: 10.3390/v15020585. PMID: 36851799; PMCID: PMC9962645.

Lesbon JCC, Poleti MD, de Mattos Oliveira EC, Patané JSL, Clemente LG, Viala VL, et al. Nucleocapsid (N) Gene Mutations of SARS-CoV-2 Can Affect Real-Time RT-PCR Diagnostic and Impact False-Negative Results. Viruses. 2021 Dec 10;13(12):2474. doi: 10.3390/v13122474. Erratum in: Viruses. 2022 Sep 05;14(9): PMID: 34960743; PMCID: PMC8707239.

Round 2

Reviewer 1 Report

Comments and Suggestions for Authors

I have no further comments. I recommend for publication. 

Author Response

We thank Reviewer #1 for his/her valuable comments that permitted considerately improve our manuscript.  We are grateful for recommending our manuscript for publication.

Reviewer 2 Report

Comments and Suggestions for Authors

Dear authors,

Thank you very much for your extensive explanations and corrections to the manuscript. It is much improved and will significantly contribute to regional and international surveillance and epidemiological efforts. Congratulations on your work.

Comments on the Quality of English Language

English is fine.

Author Response

We greatly appreciated the reviewer#2 comments, which have led to considerable improvement of the manuscript. We have revised the English language and changes in the text have been marked in red color.

We thank Reviewer#2 for the positive feedback on our manuscript.